# An Effective and Robust Approach Based on R-CNN+LSTM Model and NCAR Feature Selection for Ophthalmological Disease Detection from Fundus Images

**DOI:** 10.3390/jpm11121276

**Published:** 2021-12-02

**Authors:** Fatih Demir, Burak Taşcı

**Affiliations:** Vocational School of Technical Sciences, Firat University, Elazig 23119, Turkey; fatihdemir@firat.edu.tr

**Keywords:** ophthalmological disease, fundus images, R-CNN+LSTM, NCAR feature selection

## Abstract

Changes in and around anatomical structures such as blood vessels, optic disc, fovea, and macula can lead to ophthalmological diseases such as diabetic retinopathy, glaucoma, age-related macular degeneration (AMD), myopia, hypertension, and cataracts. If these diseases are not diagnosed early, they may cause partial or complete loss of vision in patients. Fundus imaging is the primary method used to diagnose ophthalmologic diseases. In this study, a powerful R-CNN+LSTM-based approach is proposed that automatically detects eight different ophthalmologic diseases from fundus images. Deep features were extracted from fundus images with the proposed R-CNN+LSTM structure. Among the deep features extracted, those with high representative power were selected with an approach called NCAR, which is a multilevel feature selection algorithm. In the classification phase, the SVM algorithm, which is a powerful classifier, was used. The proposed approach is evaluated on the eight-class ODIR dataset. The accuracy (main metric), sensitivity, specificity, and precision metrics were used for the performance evaluation of the proposed approach. Besides, the performance of the proposed approach was compared with the existing approaches using the ODIR dataset.

## 1. Introduction

The retina is the network layer that contains light-sensitive cells and nerve fibers and carries out vision. Lesions on the retina indicate different ophthalmological diseases such as diabetic retinopathy, AMD, cataracts, myopia, glaucoma, and hypertension. If these lesions are not examined in the early period and the related disease is not treated, partial or complete loss of vision may occur in some cases [1,2,3]. Therefore, the examination of retinal tissue is very important for a person’s eye health. Ophthalmoscope, Fundus Camera, Scanning Laser Ophthalmoscope (SLO), and Optical Coherence Tomography (OCT) devices are used for retinal imaging. Different scanning methods such as Fundus imaging, Fundus Fluorescein Angiography (FFA), and Indocyanine Green Angiography (ICG) utilize these devices. Among these methods, fundus imaging is frequently utilized since it is a noninvasive and low-cost technique [4]. Fundus imaging provides a color display of the optic nerve, macula, retina, blood vessel, and the structures of the bottom of the eye such as the vitreous. Specialists and physicians determine the ophthalmological diseases with patient anamnesis and tests based on extensive observation using fundus images. Physicians who do not have sufficient clinical experience may make incorrect decisions during the diagnosis process when their excessive workload is taken into account. Computer-aided systems can automatically detect ophthalmological diseases and make a significant contribution to the decision-making process of physicians. Especially, studies based on deep learning, which is a subfield of machine learning, have achieved high performance in classification tasks of medical images.

In this study, a robust and effective approach based on the R-CNN+LSTM was presented for automated ophthalmological disease detection from fundus images. The proposed approach was evaluated on the ODIR dataset and outperformed other existing approaches using the same dataset in several metrics. The contributions and limitation of the proposed approach can be expressed as follows.

Contributions:With the proposed R-CNN+LSTM, R-CNN and LSTM structures were trained together. Thus, the residual layer information of the R-CNN model and the LSTM model’s ability to keep important data in memory was utilized.The residual strategy and LSTM structure of the proposed R-CNN+LSTM boosted the classification achievement.The NCAR feature selection algorithm based on the calculation of feature importance and weights improved the classification performance. Besides, the NCAR algorithm, which benefited from the NCA and ReliefF algorithms, outperformed both algorithms that were popular in the selection based on feature importance and weights.

Limitation:The proposed R-CNN+LSTM model contains too many learnable parameters. Therefore, powerful hardware is required for fast prediction results.

## 2. Related Works

Several approaches based on classical machine learning and deep learning techniques have been conducted for detecting ophthalmologic disease. Almazroa et al. [5] applied a segmentation methodology to find disc and cup boundaries in glaucoma. In the classification stage, Support Vector Machine (SVM), (K-Nearest Neighbors) KNN, and Bayesian algorithms were then executed to determine 15 normal and 21 glaucomatous images. For normal and abnormal images, success rates were 100.0% and 95.23% in the SVM, 93.3% and 80.9% in the KNN, and 86.6% and 95.23% in the Bayesian, respectively. Reza and Eswaran [6] automatically detected two-class fundus images, normal and abnormal, using a rule-based classifier. In the proposed system, fundus images were preprocessed using morphological and thresholding-based techniques to remove abnormal signs such as hard exudates and cotton wool spots. In this study, DR samples were detected with an average accuracy of 97.0%. Ashraf et al. [7] used Local Binary Patterns (LBP) for the detection of hemorrhages and microaneurysms (HMAs) in the feature extraction process. The SVM with ROIs was utilized to see if samples contained HMAs. The proposed method reached 85.99% specificity, 87.48% sensitivity, 0.87 AUC, and 86.15% average accuracy for a binary classification task. Deep learning approaches have been popular in the research community and have mostly provided high performance for medical image classification tasks after the CNN model proposed by Krizhevsky et al. [8] was presented at the ImageNet Challenge in 2012 [9,10,11,12,13,14,15]. Orfao and Haar [16] operated different pretrained models such as InceptionV3, Alexnet, VGGNet, and ResNet for detecting Glaucoma, Diabetic Retinopathy, and Cataracts from fundus images. The best performance was achieved by the InceptionV3 model with an accuracy of 99.30% and an F1-Score of 99.39%. Yaroub Elloumi [17] constituted a high-performance cataract grading method with a low computational cost for smartphones. Firstly, deep features were extracted through the MobileNet-V2 model using transfer learning. Cataract grades were detected with a random forest classifier that used deep features. The best performances for specificity, sensitivity, precision, and accuracy metrics were 89.58%, 91.43%, 92.75%, and 90.68%, respectively. Khan et al. [18] utilized the average values of the predictions achieved from pretrained CNN models containing ResNet50, InceptionResNetV2, EfficientNetB0, and EfficientNetB2 in the transfer learning pipeline to improve the classification performance. In the study, an enhancement and adaptive histogram equalization technique based on morphological operations was used instead of raw images. For binary classification, the proposed ensemble-based approach outperformed the pretrained CNN models. The accuracy scores of the ResNet50, EfficientNetB0, EfficientNetB2, InceptionResNetV2, and ensemble models were 82.57%, 80.63%, 81.67%, 84.22%, and 86.08%, respectively. Khan et al. [19] opted for a structure based on the VGG19 model to detect cataracts automatically from color fundus images; 97.47% accuracy and 97.47% prediction were achieved with this model. Sun and Oruc [20] tried to diagnose ophthalmological diseases containing cataract, glaucoma, pathological myopia, hypertensive retinopathy, AMD degeneration, and diabetic retinopathy classes using transfer learning with the ResNet50. The accuracy results for cataract, glaucoma, pathological myopia, hypertensive retinopathy, AMD degeneration, and diabetic retinopathy classes were 94.9%, 89.7%, 87.0%, 93.8%, 90.8%, and 78.9%, respectively. Li et al. [21] obtained a sensitivity of 98.6% in the classification of AMD and DME using the VGG16 model on a dataset containing 207,130 images taken through OCT. Raghavendra et al. [22] developed an eighteen-layer convolutional neural network for the diagnosis of glaucoma from fundus images. With this developed model, an accuracy rate of 98.13% was achieved. Singh et al. [23] designed a lightweight CNN model for the detection of the DR disease and the classification of DR disease stages (5 classes). The successes of the study were 71% for two classes and 56% for five classes. Chai et al. [24] proposed a multibranch neural network model containing faster R-CNN, fully convolutional network (FCN), and custom CNN models for the detection of glaucoma. By testing the proposed model on the dataset, a success of 91.51% was achieved.

## 3. Methodology, Material, and Techniques

### 3.1. Proposed Methodology

The framework of the proposed approach is given in Figure 1. In this study, a novel approach was proposed for automated ophthalmological disease detection from fundus images. The proposed approach was composed of four steps. In the first step, the proposed R-CNN+LSTM was trained on the dataset. The residual strategy and the LSTM model containing 100 LSTM units were used for boosting classification performance. The representation of the R-CNN+LSTM model consisting of six residual blocks is shown in Figure 2. 

Each residual block was constituted of two convolutional units, two BN layers, and a ReLU layer. The filter weights and activations of the R-CNN were conveyed to the unfolding layer for the learning process together with the LSTM model. Since important information was stored in the LSTM structure, it was used with the R-CNN structure and the classification performance was increased. However, it cannot be said that the softmax classifier used in deep-learning-based approaches with an end-to-end learning strategy will give the best performance for every classification task. Therefore, in the second step, for boosting classifier performance, other robust classifier algorithms such as SVM, K-NN, and Decision Tree instead of the softmax classifier were evaluated with the trained activation values of the R-CNN+LSTM. Therefore, deep features were extracted from the first fully connected layer output of the R-CNN+LSTM model, which included an end-to-end learning process. In the third step, distinctive features were selected using the NCAR algorithm that had a multilevel selection strategy with the NCA and ReliefF algorithms. With this algorithm, the classification achievement was improved and the computational cost of the classifier was reduced. In the fourth step, the selected features were transmitted to the SVM classifier. The SVM classifier was evaluated on the dataset with 10-fold cross-validation.

### 3.2. Dataset

The ODIR dataset consisted of color fundus images collected from the left and right eyes of volunteer patients [25]. Fundus images were constituted by various cameras in the market such as Canon, Zeiss, and Kowa and were then saved in different sizes and dpi values in JPG format. The dataset included 3098 Normal, 1406 Diabetes, 224 Glaucoma, 265 Cataract, 293 AMD, 107 hypertension, 242 Pathological Myopia (PM), and 791 other diseases/abnormalities in total; the 8 classes comprised 6426 samples. All classification processes in the dataset were performed by expert ophthalmologists. Fundus images were rerecorded in JPG format and standard sizes (125 × 125) with 96 dpi. In the ODIR dataset, some examples for each class are given in Figure 3.

### 3.3. Deep Learning Techniques

The purpose of the convolution layer in CNN is to extract distinctive information by processing input samples with convolution filters. Convolution is a mathematical operation of two functions. In the CNN concept, the convolution operation simply shifts a kernel function, also called a filter, over the master data by performing the element-wise multiplication of each element. For each window in the shift operation, the sum of the multiplication with element information gives the result for that window. By scrolling windows across the entire image, the output of the convolution operation called the feature map is produced. During the network design, there are three hyperparameters to be selected for the convolution layer. These are the dimensions of the convolution filter, the step size of the convolution filter while hovering over the input image, and whether any padding will be applied to the input image [8,26].

Batch normalization (BN) is a method used to make the convolutional neural network more regular. Besides a regulatory effect, it also gives resistance to the extinction gradient of the convolutional neural network during training. In short, BN is a method that increases the speed, performance, and continuity of deep neural networks [27,28].
(1)$${µ}_{b}=\frac{1}{n}\overset{n}{\underset{i=1}{\sum }}x_{i}$$
(2)$$\sigma_{b}=\frac{1}{n}\overset{n}{\underset{i=1}{\sum }}{\left(x_{i}-{µ}_{b}\right)}^{2}$$
(3)$${\hat{x}}_{i}=\frac{x_{i}-{µ}_{b}}{\sqrt{\sigma_{b}^{2}+\varepsilon }}$$
(4)$$y_{i}=\alpha {\hat{x}}_{i}+\beta$$

LSTM is a special type of RNN with the ability to learn long-term dependencies. This model, which was first proposed in the mid-90s, is widely used today [29]. Although it is aimed to store and transfer the state information of the artificial neural network while processing the sequences in RNNs, it is not possible to transfer the state information without spoiling the long-term dependencies as a result of the continuous processing of the state information. In other words, while short-term addictions in the series are transferred quite successfully, there is a problem in transferring long-term addictions. The basic principle behind this network architecture is that the network reliably transmits important information into the future in multiple iterations [30]. The LSTM memory cell is given in Figure 4. There are 3 doors in LSTM. These are the entrance, forget, and exit doors. These gates in LSTM are sigmoid activation functions. In Equations (5)–(10), where *W* denotes the weight matrices, *Ct* is the cell state, *b* is the input bias vector, *i* represents the input gate, *f* stands for the forget gate, and ot symbolizes the output gate. The extracellular activation function is tanh. The output layer is the last layer in the network used to estimate the sensitivity. The basic LSTM architecture consists of input (Equation (6)), output (Equation (9)), forget gates (Equation (5)), and memory cells (Equation (3)).
(5)$$f_{t}=\sigma (W_{f}\cdot [h_{t-1},x_{t}]+b_{t})$$
(6)$$i_{t}=\sigma (W_{i}\cdot [h_{t-1},x_{t}]+b_{i})$$
(7)$${\tilde{C}}_{t}=\tanh(W_{c}\cdot [h_{t-1},x_{t}]+b_{c})$$
(8)$$C_{t}=f_{t}*C_{t-1}+i_{t}*{\tilde{C}}_{t}$$
(9)$$o_{t}=\sigma (W_{o}\cdot [h_{t-1},x_{t}]+b_{o})$$
(10)$$h_{t}=o_{t}*\tanh(C_{t})$$

The ReLU layer is the layer that applies an activation function *f*(*x*) = max(0, *x*) to each element in its input. ReLU, which is a nonlinear activation function, sets its less-than and equal inputs to zero while leaving its greater than zero inputs as they are. In CNN models, the ReLU layer is used after the convolution layers. The ReLU layer is applied one by one for each element of the input and sets values less than 0 to 0 while leaving values greater than 0 as they are. The use of the ReLU function is preferred because it is several times faster than other activation functions such as sigmoid or hyperbolic tangent, although it does not make a significant difference in generalization accuracy. This difference provides great ease of application in deep artificial neural networks where the computational load is quite high [31,32]. As can be seen in Equations (11) and (12), the fact that its derivative is simpler than the sigmoid function provides great convenience and speed when using algorithms such as backpropagation.
(11)$$f\left(x\right)=\max(0,\;x)=f^{\prime }\left(x\right)=\{\begin{array}{c} x>0\to 1\\ x\leq 0\to 0 \end{array}$$
(12)$$\sigma \left(z\right)=\frac{1}{1+e^{-z}}\;\;{\sigma^{\prime }}^{\left(z\right)}=\sigma \left(z\right)\left(1-\sigma \left(z\right)\right)$$

The flattening layer is the conversion of a two-dimensional feature matrix into a one-dimensional vector to feed the next layer [33].

The softmax function is often used in the output of deep learning models. The softmax function sets the class scores generated in the fully connected layer to probability-based values between 0 and 1. The softmax function *s*(*a_j_*)takes an N-dimensional input vector, as seen in Equations (13) and (14), and produces a second N-dimensional vector with each element having values between 0 and 1. Although the softmax function is generally used in the output layer of deep learning models, a classifier such as the support vector machine (SVM) can also be used. Since it is an exponential function, the softmax function makes the difference between classes even more pronounced.
(13)$$S\left(a_{j}\right):\left[\begin{array}{c} a_{1}\\ \ldots .\\ a_{n} \end{array}\right]\to \left[\begin{array}{c} s_{1}\\ \ldots .\\ s_{n} \end{array}\right]$$
(14)$$S\left(a_{j}\right)=\frac{e^{a_{j}}}{\sum_{k=1}^{n}e^{a_{k}}}$$

The dropout layer is used to forget some neurons to avoid overfitting during training. There is a risk of overlearning in cases where the network structure is large, when training is done for too long, or when the number of data is too small [34]. 

### 3.4. Multilevel Feature Selection

Feature selection methods aim to improve execution velocity without reducing approach achievement. In community research, many feature selection algorithms have been used for the machine learning approach. Especially, feature selection techniques reduce execution time in deep learning applications having many features. To determine which feature selection algorithm will provide good performance on which feature set, the data in the feature set should be analyzed well. However, it is quite burdensome to apply this analysis process, especially in deep-learning-based approaches. For example, LDA and PCA algorithms perform well on a linear feature set only, while the mRMR algorithm performs well on a nonparametric feature set. In recent studies, feature-importance-based selection algorithms have been started to select for classification problems [35,36,37]. The most popular feature-importance-based selections algorithms are the NCA and ReliefF since they provide various classification algorithms. Besides, the execution time of feature selector for these algorithms are lesser than algorithms such as PCA and mRMR.

In this study, a multilevel feature selection method named NCAR, containing the NCA and ReliefF algorithms, was used for the proposed approach for boosting classification performance. For computing feature importance weights, the ReliefF algorithm utilizes distance and a nearest-neighbor-based technique while the NCA algorithm utilizes distance with kernel and a probability-based technique. Thus, the representation powers of the two algorithms were benefited in the feature selection process. The pseudocode of the NCAR is expressed in the Algorithm 1.


**Algorithm 1. Pseudocode of the NCAR algorithm**
Input: feature vector from CovEncoNet model (fea), size of feature vector (N)  average of fea(avg), standard deviation of fea (std), threshold (thr)Output: reduced feture vector (fea_out)*1:. feature_reducion(fea,std,avg,thr)**2:. begin**2:. fea_out = fea**3:. for i = 1 to N do**4:. decision1 = std/fea_out[i]**5:. decision2 = avg/fea_out[i]**6:. if decision1 > thr and decision2 > thr**7:. fea_out[i] = []**8:. end if**9:. end for i**10:. end*

Neighborhood Component Analysis (NCA) is a dimension reduction, feature selection technique. The measurement of features is very important in machine learning applications. One of the most successful learning algorithms, NCA, is widely used in classification studies [33]. Neighborhood component analysis performs classification operations by learning the projection of the vectors that optimize the criteria concerned with the classification accuracy of the nearest neighbor classifier. In other words, the NCA chooses a linear projection that optimizes the performance of the nearest neighbor classifier in the projected area. NCA uses training data consisting of associated class labels when choosing the projection that will be effective in separating classes in the prescribed area. NCA makes weak assumptions about the distribution in each class when optimizing its classifiers. This gives a closer match to the use of Gaussian mixtures in modeling distributions in classes [34]. The regularized objective function [38] given in Equation (15) is used. Thus, in the NCA method, the aim is to maximize the objective function *F*(*w*) for *w*.
(15)$$F\left(w\right)=\frac{1}{n}\overset{n}{\underset{i=1}{\sum }}P_{i}-\lambda \overset{p}{\underset{r=1}{\sum }}w_{r}^{2}$$
here, λ is the regularization parameter, *p* is its dimensionality, *w_r_* is the feature weight, n is the total number of samples, and *P_i_* represents the probability score of *i*th sample. When the λ parameter is chosen randomly, all feature weights can take values very close to zero. The fact that the weights are close to zero in this method indicates that the relevant features are unimportant. Therefore, the parameter λ needs to be adjusted.

ReliefF algorithm is an algorithm that can make effective feature predictions. These feature estimates are made by using the feature weights. The feature weights are determined by solving the convex optimization problem [39]. Firstly, the weights of all features are set to 0. Then, at each step, it randomly selects data from the data set and finds the closest *k* (*k* value is one less than the number of classes) data belonging to the same class with this data, and then the closest data belonging to each different class are found. Then, the weights of each feature are updated using this data. At the last stage, the features that do not meet the specified condition are removed from the data set, and a new data set is created. The ReliefF algorithm was formulated as follows.
(16)$$W(x^{a})=W(x^{a})-\frac{\sum_{j=1}^{k}diff(A,R_{i},H_{j})}{mxk}+\underset{c\neq class\left(R_{i}\right)}{\sum }\left[\frac{P\left(C\right)}{1-P(class\left(R_{i}\right)}x\overset{k}{\underset{j=1}{\sum }}\frac{diff\left(A,R_{i,M_{j}}\right)}{mxk}\right]$$
here, *x^a^* represents the *a*th feature, *A* is the feature set, *R_i_* and *H_j_* stand for instances in the feature set, and *m* and *k* symbolize the user-selected parameter.

## 4. Experimental Studies

The algorithm related to the proposed approach was operated using MATLAB software installed on the Windows 10 operating system and hardware containing an i7 Intel Core ™ processor, 8 GB RAM, and 2 GB graphic card. The mini-batch size, initial learning rate, and max epochs adjusted as training option parameters of the proposed R-CNN-LSTM model were set to 128, 0.001, and 150, respectively. The SGDM was used as optimization solver since CNN models mostly provided good performance. Besides, the cross-entropy was selected as the loss function of the R-CNN+LSTM model. 

In Figure 5, the accuracy and loss graphs of the proposed R-CNN+LSTM model are given during the training process. At the end of 2000 iterations, the accuracy and loss scores of training were 100.0% and 0.0250, respectively. A total 350 features were extracted using the activation values of the first fully connected layer in the trained R-CNN+LSTM model. Then, distinctive features (38 features) were selected by the NCAR algorithm. In the first level of the NCAR algorithm, the feature weights shown in Figure 6 were computed with the NCA algorithm. A total 292 features, which were less than the selected threshold value (0.0005) for features weight value, were removed from the feature set. In the second level of the NCAR algorithm, the feature importance weights were calculated with the ReliefF algorithm and the number of nearest neighbors was selected as 10. As seen in Figure 7, by using the threshold with a feature importance weight of 0.01, 30 features were selected from 58 features. 

According to all classes and the levels of the NCAR feature selection algorithm, 3D representations of features sets are given for a sample in Figure 8. In Figure 8, the first, second, and third columns show the features without feature selection operation, the features selected with the first level of the NCAR algorithm, and the features selected with the second level of the NCAR algorithm, respectively. As seen in Figure 8, the features in the feature set obtained by the NCAR algorithm are morphologically better differentiated from the raw deep features (350 features).

In Table 1, the accuracy results containing three different feature selection cases (NCA, ReliefF, and NCAR) are given for Decision Tree (DT), Linear Discriminant (LD), Naïve Bayes (NB), K-Nearest Neighbors (KNN), and SVM classifiers. For all feature selection algorithms, the number of the selected features was adjusted as 30. As seen in Table 1, the best accuracy was 89.54% with the SVM classifier and the NCAR algorithm while the worst accuracy was obtained as 75.85% with the NB classifier and the NCA algorithm. Among all classifiers, the best performances for all feature selection algorithms were achieved with the SVM classifier. The best performance of classifiers in order was SVM, KNN, DT, LD, and NB. Among feature selection algorithms, the best performance for all classifiers was achieved with the NCAR feature selection algorithm. The best performance order of feature selection algorithms was NCAR, ReliefF, and NCA.

In Figure 9 and Figure 10, the confusion matrices and the ROC curves with AUC values are given for six different cases, respectively. In the first case, the CNN structure in the proposed approach was used without residual blocks. In the second case, the CNN+LSTM structure in the proposed approach was used without residual blocks. In the third case, the R-CNN structure in the proposed approach was used without the LSTM model. In the fourth case, the proposed CNN+LSTM structure was used. In the fifth case, the R-CNN+LSTM+SVM structure in the proposed approach was used without the NCAR feature selection algorithm. In the sixth case, the R-CNN+LSTM+SVM structure in the proposed approach was used with residual blocks. As seen in Figure 9, the classification accuracy with adding the LSTM model and the residual blocks to the CNN structure was improved by 0.92% and 4.28%, respectively. Instead of the fully connected + softmax classifier in the proposed R-CNN+LSTM, using the SVM classifier increased the accuracy score by 1.66% (the fifth case). With the NCAR feature selection algorithm (sixth case), the classification performance of the proposed approach was improved by 0.28% compared with the fifth case without feature selection. As seen in Figure 10, the worst and best AUC values were obtained as 0.85 and 0.97 with the CNN structure (the first case) and the proposed approach (the sixth case), respectively.

In Table 2, the sensitivity, specificity, precision, and F-score results are given for all classes of the proposed approach. The best sensitivity was obtained as 0.9777 with the Normal class and the worst sensitivity was obtained as 0.8020 with the AMD class. The best specificity was achieved as 1.0 with the Glaucoma class and the worst specificity was achieved as 0.8275 with the Normal class. The best precision was obtained as 1.0 with the Glaucoma class and the worst precision was obtained as 0.8421 with the Normal class. The best F-score was 0.9341 with the Hypertension class and the worst specificity was 0.8764 with the Other class.

The SVM accuracy performances of features extracted with the proposed R+CNN+LSTM and the other CNN backbones are given in Table 3. As seen in Table 3, the NCAR feature selection strategy improved the classification achievement of all CNN backbones.

## 5. Discussion

Many deep-learning-based studies have been conducted in the research community using fundus images. Since these studies are carried out on different data sets and different training parameters are used in the proposed approaches, one method cannot be said to be completely superior to the others. Besides, when these studies are examined in general, it is more difficult to achieve performance in multiclass classification tasks than in two-class classification tasks. Therefore, the proposed approach was evaluated on the ODIR dataset containing eight classes. 

For the ODIR dataset, the AUC and F-score results of the proposed approach and the existing approaches are given in Table 4. Islam et al. [40] proposed a lightweight CNN model trained from scratch for automated ophthalmological disease from fundus images. This approach reached an AUC of 80.50% and an F-score of 85.00%. Jordi et al. [41] and Li et al. [42] presented transfer learning approaches. Jordi et al. [41] achieved 88.71% AUC and 81.76% F-score with the VGG16 model and Li et al. [42] obtained 93.00% AUC and 91.30% F-score with the ResNet101 model. He et al. [43], utilized the pretrained CNN models containing ResNet18, ResNet34, ResNet50, and ResNet101 for the classification task., With the ResNet101 model, the best F1 Score and AUC results were obtained as 90.70% and 92.70%, respectively. Wang et al. [44] presented a novel model named EfficientB3 consisting of two models. The EfficientNet model and the processed images with gray histogram equalization were utilized in the first model while The EfficientNet model and the processed images with color histogram equalization were utilized in the second model. The prediction results of these models were combined with the majority vote technique for boosting the classification performance. The EfficientNetB3 model provided an accuracy of 89.00%, an AUC of 73.00%, and an F1 score of 89.00%. Gour and Khanna [45] utilized four pretrained CNN models containing ResNet, InceptionV3, MobileNet, and VGG16. The best AUC and F-score results were 84.93 and 85.57, respectively. The proposed approach provided the best AUC score with 97.00%. However, the best F-score value was achieved with the approach proposed by Li et al. [42]. The second-highest F-score was obtained by the proposed approach. However, the CNN models in [41,42,43,44,45] are pretrained models. Since the weights of these models are shared, no training is required for deep feature extraction. Further, in the CNN model proposed in [40], fewer layers were used compared with the proposed R-CNN+LSTM. Therefore, the computation speed of the proposed method is lower than the existing methods in hardware with the same capacity.

Among the existing approaches, only the approach proposed by Gour and Khanna [45] yielded the results of sensitivity, specificity, and accuracy metrics. In Table 5, according to the performance metrics of accuracy, AUC, and F-score, the proposed approach is compared with the pretrained CNN models (the ResNet, InceptionV3, MobileNet, EfficientB3, and VGG16 models) used by Gour and Khanna [45]. As seen in Table 5, for all metrics, the best performance was obtained with the proposed approach while the worst performance was obtained with the ResNet model in [45].

For all classes, the sensitivity and specificity results of the proposed method and the VGG16 model (the pretrained model having the best performance in [45]) are given in Table 6. As seen in Table 6, the results of the proposed approach for both sensitivity and specificity are more balanced when examined in general. For the sensitivity metric, the proposed approach provided better performance in the Glaucoma, Hypertension, Normal, and Other Disease classes. Especially, the performance for the Normal class was improved at a high rate (0.3177). For all remaining classes, the VGG16 model outperformed the proposed approach. In the specificity metric, for the Cataract class, the VGG16 model in [45] outperformed the proposed approach by a little margin (0.01). The sensitivity scores were improved for all remaining classes except the Hypertension class. Especially, the performances for the Normal and Other classes were improved at high rates (0.61 and 0.67).

## 6. Conclusions

In this study, a novel and robust approach was proposed for automated ophthalmological disease detection from fundus images. The proposed approach was evaluated on the eight-class ODIR dataset. In the proposed approach, the R-CNN+LSTM architecture was used to extract deep features. Using residual strategy and adding the LSTM model in the R-CNN+LSTM model, the classification accuracy improved by 4.28% and 1.61%, respectively. For obtaining the highest accuracy and reducing the classifier execution time, a multilevel feature selection algorithm named NCAR was applied to 350 deep features. Using the DT, NB, LD, SVM, and KNN classifiers, the performance of the NCAR was compared with the NCA and ReliefF algorithms. The best accuracy was achieved with the NCAR feature selection algorithm and the SVM classifier. For the proposed approach, the best accuracy was obtained as 89.54% and the classification accuracy was improved by 0.28%. Besides, the proposed approach was compared with the existing approaches using the ODIR dataset. With the proposed approach, the AUC and accuracy values were improved by 4% and 0.48%, respectively. The proposed approach for the F-score metric reached the third-best value with 0.8994 (the best value was 0.9134 and the second-best value was 0.9070). Moreover, according to accuracy, AUC, and F-score metrics, the proposed approach was compared with pretrained CNN models in [45] and the EfficientB3 model in [44]. The proposed approach outperformed these pretrained CNN models. Moreover, for each class according to sensitivity and specificity metrics, the proposed approach was compared with the pretrained CNN model (VGG16) providing the best performance in [45]. In four classes for sensitivity and six classes for specificity, the proposed approach outperformed the VGG16-model-based approach. However, robust hardware is required for the proposed approach based on the deep learning strategy. With more powerful hardware in the future, it is considered that the performance evaluation should be repeated by adding attention structures to the proposed approach.

## Figures and Tables

**Figure 1 jpm-11-01276-f001:**
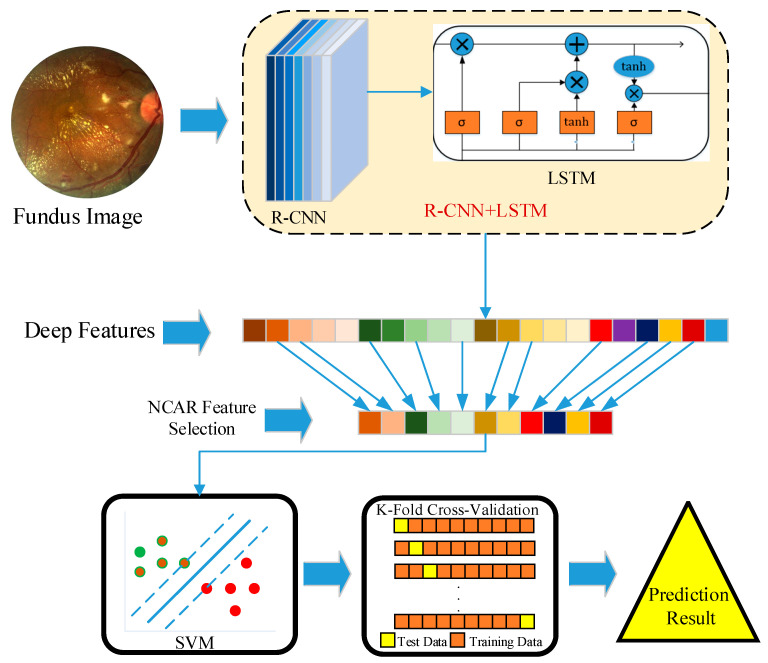
Framework of the proposed approach.

**Figure 2 jpm-11-01276-f002:**
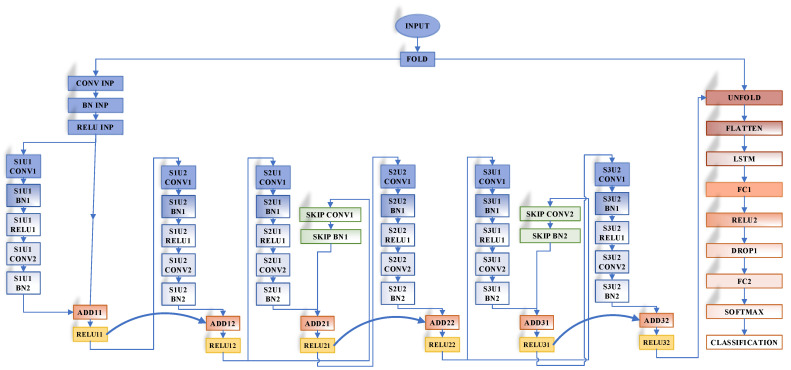
Representation of the proposed R-CNN+LSTM.

**Figure 3 jpm-11-01276-f003:**
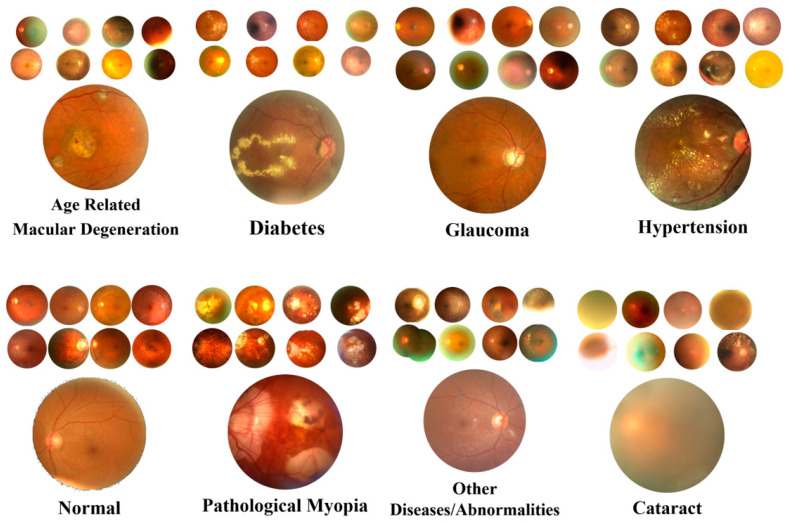
Some samples for each class in the dataset.

**Figure 4 jpm-11-01276-f004:**
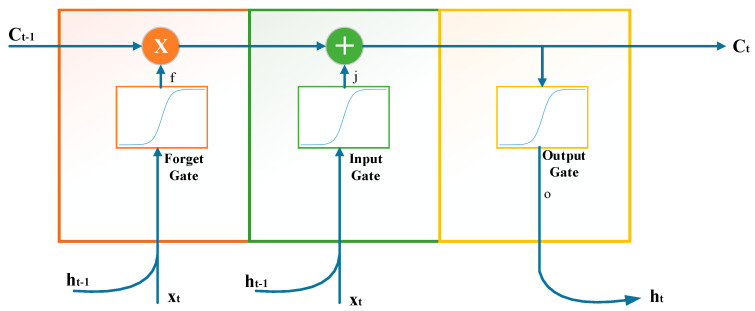
An LSTM Memory Cell.

**Figure 5 jpm-11-01276-f005:**
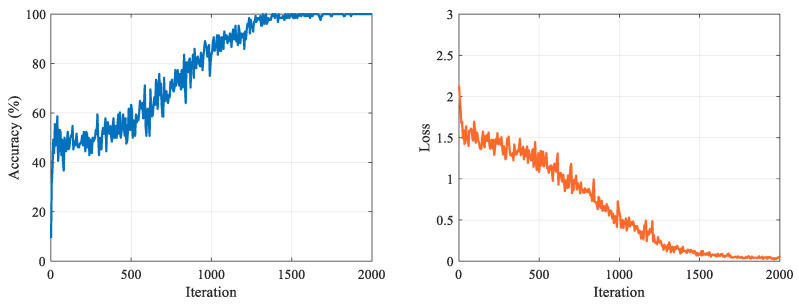
Training accuracy and loss graphs of the R-CNN+LSTM model.

**Figure 6 jpm-11-01276-f006:**
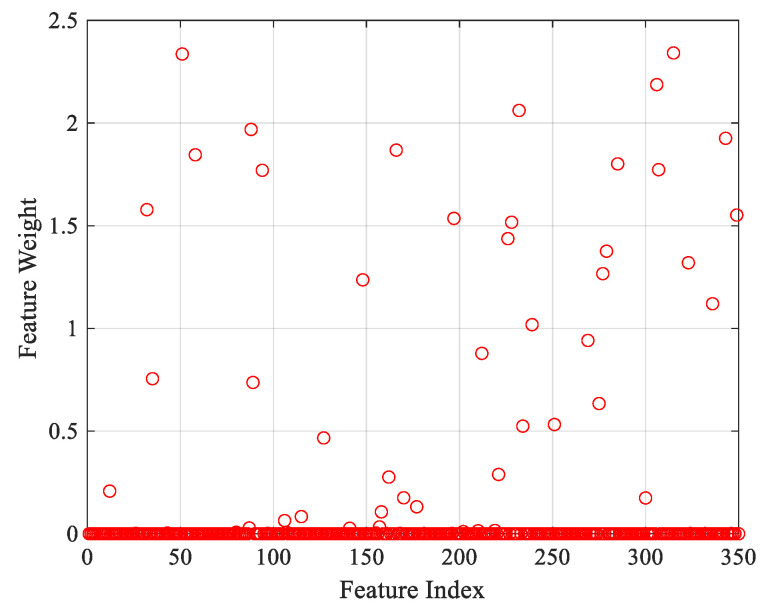
Deep feature weights for each feature index.

**Figure 7 jpm-11-01276-f007:**
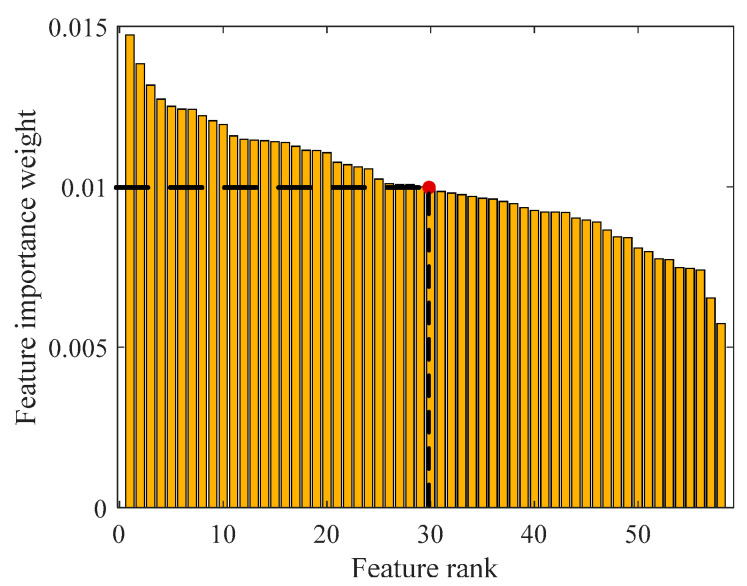
Training accuracy and loss graphs of the R-CNN+LSTM model.

**Figure 8 jpm-11-01276-f008:**
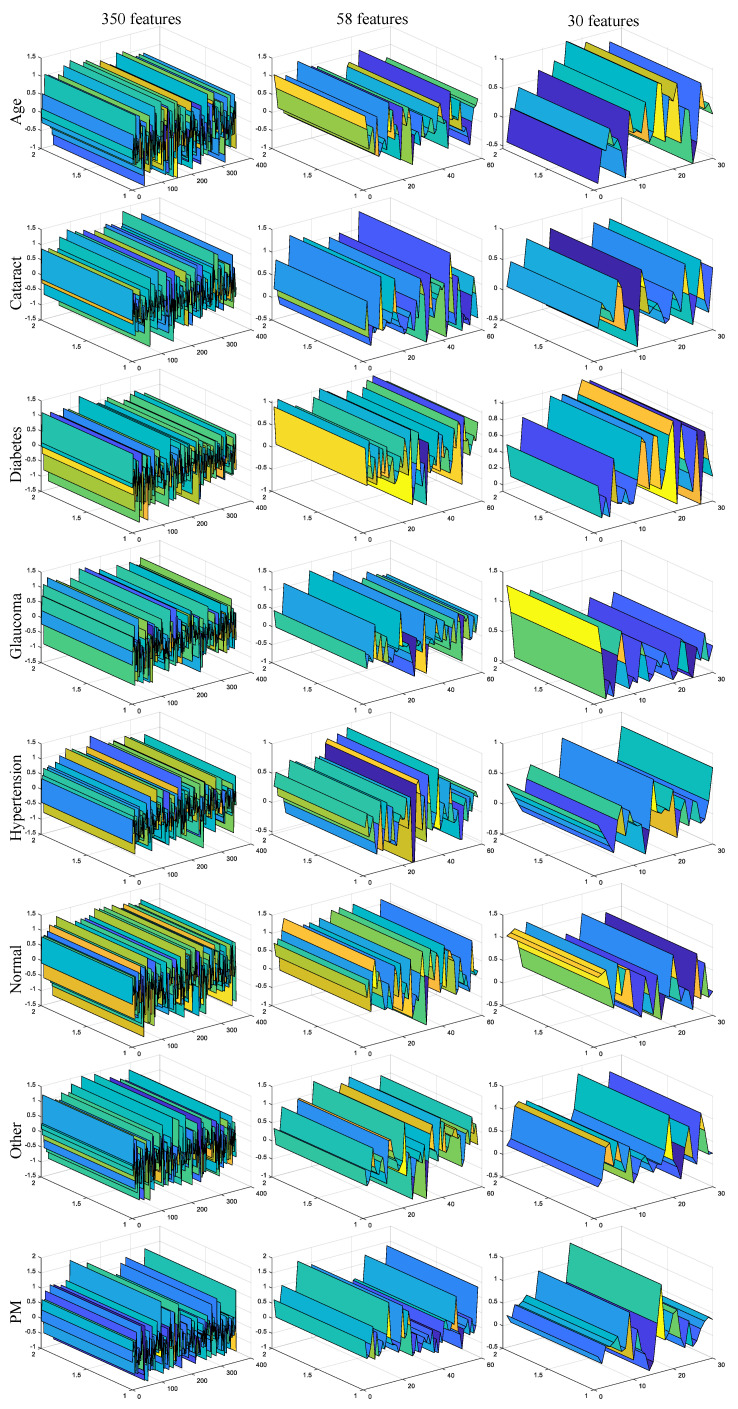
3D representations of deep features for feature selection situations.

**Figure 9 jpm-11-01276-f009:**
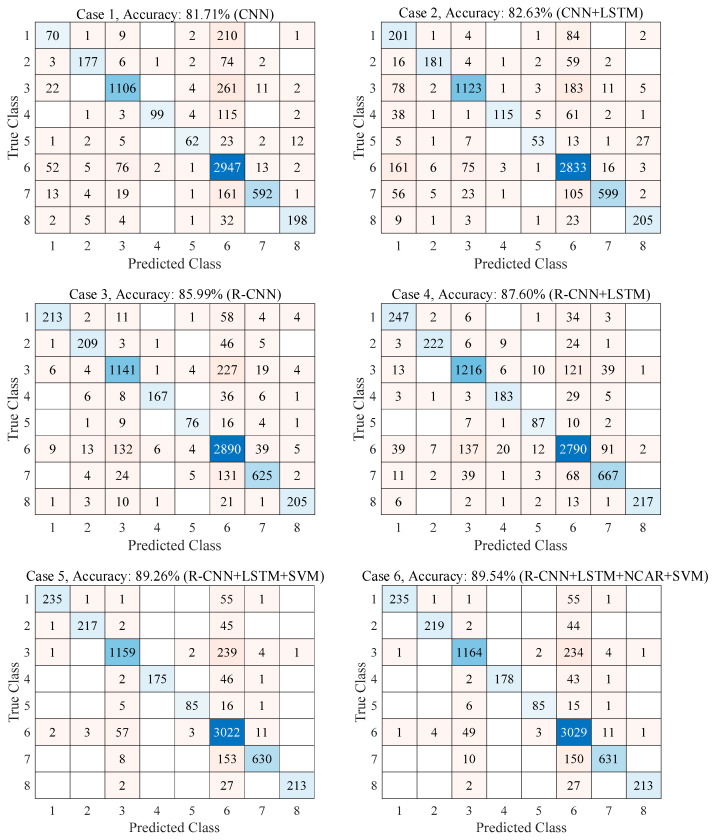
Confusion matrices for six different approaches (**1**: AMD, **2**: Cataract, **3**: Diabetes, **4**: Glaucoma, **5**: Hypertension, **6**: Normal, **7**: Other Disease, **8**: PM).

**Figure 10 jpm-11-01276-f010:**
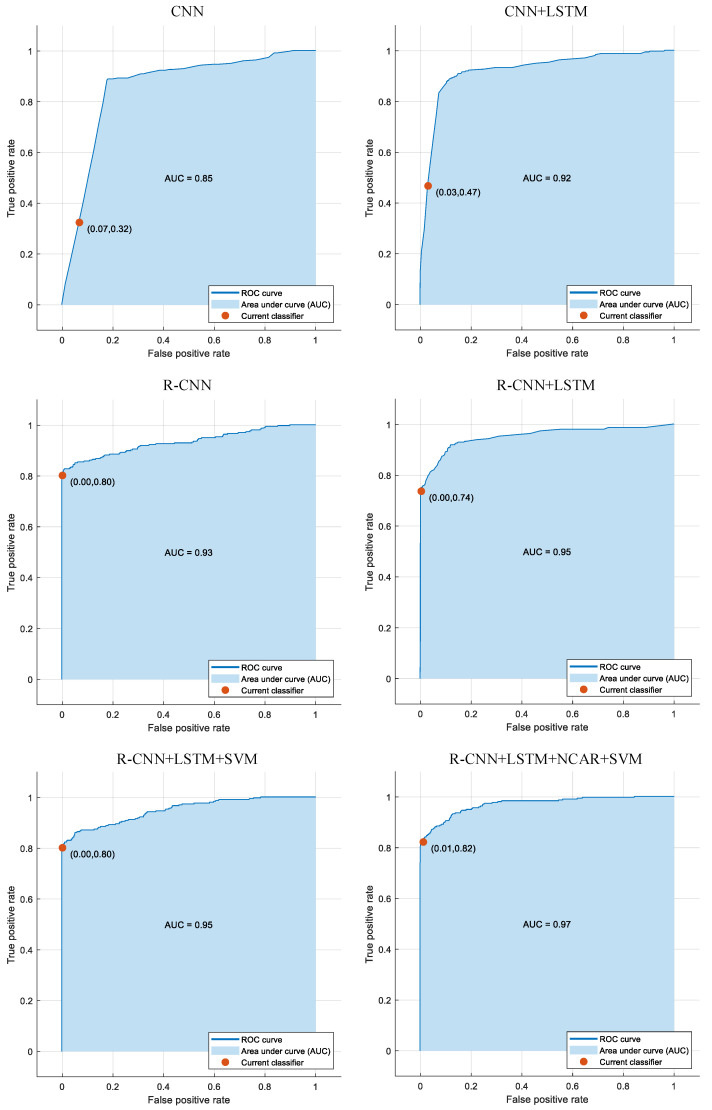
The ROC curves and AUC values for six different approaches.

**Table 1 jpm-11-01276-t001:** The classifier performance results according to feature selection algorithms.

Classifier	Accuracy (%)		
	NCA	ReliefF	NCAR
DT	80.15	81.28	81.95
LD	78.76	79.87	80.35
NB	75.85	76.34	76.94
SVM	89.28	89.35	89.54
KNN	88.56	88.94	89.34

**Table 2 jpm-11-01276-t002:** The other performance results of the proposed approach.

Class	Sensitivity	Specificity	Precision	F-Score
AMD	0.8020	0.9996	0.9916	0.8868
Cataract	0.8264	0.9991	0.9777	0.8957
Diabetes	0.8285	0.9846	0.9417	0.8815
Glaucoma	0.7946	1.0000	1.0000	0.8856
Hypertension	0.9239	0.9991	0.9444	0.9341
Normal	0.9777	0.8275	0.8421	0.9049
Other Disease	0.7977	0.9965	0.9723	0.8764
PM	0.8802	0.9996	0.9907	0.9322

**Table 3 jpm-11-01276-t003:** The other performance results of the proposed approach.

Class	No Feature Selection	NCAR Feature Selection
CNN	0.8171	0.8225
CNN+LSTM	0.8263	0.8375
R-CNN	0.8599	0.8725
R-CNN+LSTM	0.8760	0.8890
R-CNN+LSTM+SVM	0.8926	0.8954

**Table 4 jpm-11-01276-t004:** The AUC and F-score results of the proposed approach and the existing approaches.

Author	Method	AUC (%)	F-Score (%)
Islam et al. [40]	CNN	80.50	85.00
Jordi et al. [41]	VGG16	88.71	81.76
Li et al. [42]	ResNet101	93.00	91.30
Wang et al. [44]	EffifinetB3	73.00	89.00
He et al. [43]	ResNet models	92.70	90.70
Gour and Khanna [45]	Two I/P VGG16	84.93	85.57
Proposed Approach	(R-CNN+LSTM)+NCAR+SVM	97.00	89.97

**Table 5 jpm-11-01276-t005:** The other performance results of the proposed approach.

Model	Accuracy (%)	AUC (%)	F-Score (%)
ResNet [45]	85.52	71.96	84.15
InceptionV3 [45]	83.98	77.16	85.47
MobileNet [45]	85.81	71.42	85.50
EfficentB3 [44]	89.00	73.00	89.00
VGG16 [45]	89.06	84.93	85.57
Proposed Approach	89.54	97.00	89.97

**Table 6 jpm-11-01276-t006:** Comparison with the model proposed in [45] of the sensitivity and specificity results.

Class↓/Metrics→	Sensitivity		Specificity	
	Gour and Khanna [45]	Proposed Approach	Gour and Khanna [45]	Proposed Approach
AMD	0.94	0.8020	0.93	0.99
Cataract	0.96	0.8264	1.00	0.99
Diabetes	0.93	0.8285	0.94	0.98
Glaucoma	0.67	0.7946	0.60	1.00
Hypertension	0.95	0.9239	0.99	0.99
Normal	0.66	0.9777	0.21	0.82
Other Disease	0.73	0.7977	0.32	0.99
Myopia	0.94	0.8802	0.94	0.99

## Data Availability

Not applicable.

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
