# Peer review of "An Effective and Robust Approach Based on R-CNN+LSTM Model and NCAR Feature Selection for Ophthalmological Disease Detection from Fundus Images"

_jpm, 2021, doi:10.3390/jpm11121276_

Round 1

Reviewer 1 Report

The Authors present a novel method for detecting ophthalmological diseases from fundus retina images, and efforts on such an important topic in ophthalmological screening should be lauded; minor revisons are requested.

The main limitations of the study ought to be fully acknowledged, both in the Introduction (line 50) and in the Discussion/Conclusions sections.

The Introduction should be revised, as there are phrases that do not make much sense, such as “Lesions on the retina can be the cause of.. cataracts..(lines 27-29)”, and the first three references should be removed as not much pertinent general ophthalmological information is given from them.  

A major revision of the References should also be performed, for example reference 21 refers to an evaluation of brain diseases, reference 4 in Table 4 refers to an ophthalmology book and not to a model, the site in reference 25 is not correct..

Minor English revisions should be performed in the whole manuscript (examples: line 119 fungus images, line 127 with together, line 325 as seen Table, line 334 algortihms, line 382 utlized, line 385 was presented, line 386 two model…)

Table 5 caption should be revised referring also to the comparison between the two approaches.

AMD abbreviation should be used instead of age-related macular degeneration after the first time.

Reviewer 2 Report

Summery: Authors propose a RCNN+LSTM-based approach to automatically detect multiple ophthalmologic diseases from fundus images. Authors adopt RCNN+LSTM to extract the features, NCAR to select the feature and SVM to classify the features. The proposed approach is evaluated on the eight-class ODIR dataset. They get very high diagnosis performance on multiple diseases, and outperform many other commonly used methods.

Advantages: The article is well written and easy to follow. The network archtecture and the proposed components are introduced in detail. The experiments and analysis on the proposed feature selection method are sufficient. Convincing experiments are conducted to show the proposed NCAR+SVM is better than the other feature selection methods and machine learning classifiers.

Flaws:

The details of the method and some other experiments should be supplied. First, the authors did not show how they train the model in detail. Since the presented deep learning model is not end-to-end, the authors should supply the training strategy they used. Best also draw in Figure 1. The authors also should demonstrate the motivation of proposing the modules. For example, why using RCNN+LSTM in this scenario? why using feature selection + SVM but not commonly used SOFTMAX + fully connected layer? Authors should explain their motivation of using this framework and why not using common end-to-end deep learning models.

Second, the experiments in the paper cannot well support the article. The main two contributions are RCNN+LSTM and NCA+ReliefF, but there has no ablation study on these two components. Authors should supply the experiments, 1. The performance of RCNN+LSTM classified by softmax+fully connected layer. 2. The performance of NCA+ReliefF on the other neural network backbones.

Details

  1. Line 289, xa is written in a wrong type.
  2. Table 3, citation of Gour and Khanna is wrong.
  3. Line 398-404, the results in Table 4 are confusing. Are they the performance of [46] with different backbones? What is the purpose of Table 4.
  4. Why the citations of resnet, inceptionv3, mobilenet and vgg16 are all [6]?
  5. Table 3, the speed of the methods should also be compared.

Round 2

Reviewer 2 Report

Authors should explain their method more clearly before being accepted.

  1. Traing strategy should include how to train the network. For example, the loss function, the learning rate, the optimizer, etc. Did the authors pre-train the network end-to-end using fc+softmax, and replace softmax by SVM? Authors should say it more clearly.
  2. Authors still have not explained their motivation. "it cannot be said that the softmax classifier used in deep learning-based approaches with an end-to-end learning strategy will give the best performance for every classification task. Therefore..." Why the authors think fc+softmax is not the best choice for this task? I mean, before the experiments. I think the proposed NCA + SVM is not the best for every tasks, right? In what cases the proposed classifier is better than commonly used  fc+softmax?
  3. line 412-417. 414 "Also..."the sentence is not complete. The network has fewer layers but used NCA+SVM to replace fc+softmax. Is NCA+SVM faster than fc+softmax?
  4. line 356, actually softmax is not a classifier, it is only a normalization operation. fc+softmax works as the classifier.
